# A New Strategy for Sponge City Construction of Urban Roads: Combining the Traditional Functions with Landscape and Drainage

**Chengyao Wei** [1,2], **Jin Wang** [3], **Peirong Li** [4], **Bingdang Wu** [1], **Hanhan Liu** [4], **Yongbo Jiang** [1] **and Tianyin Huang** [1,2,*]

1   School of Environmental Science and Engineering, Suzhou University of Science and Technology, Suzhou 215009, China; wcydxszl@163.com (C.W.); wubingdang@163.com (B.W.); SZjiangyongbo@163.com (Y.J.)
2   Key Laboratory of Suzhou Sponge City Technology, Suzhou 215002, China
3   Housing and Urban-Rural Construction Bureau of Suzhou, Suzhou 215002, China; vae2116@163.com
4   Suzhou Tongke Engineering Consulting Co., Ltd., Suzhou 215000, China; lprkycn@126.com (P.L.); yqdhhqldy@163.com (H.L.)
*   Correspondence: huangtianyin111@163.com

**Abstract:** Urban roads play a key role in sponge city construction, especially because of their drainage functions. However, efficient methods to enhance their drainage performance are still lacking. Here, we propose a new strategy to combine roads, green spaces, and the drainage system. Generally, by considering the organization of the runoff and the construction of the drainage system (including sponge city facilities) as the core of the strategy, the drainage and traffic functions were combined. This new strategy was implemented in a pilot study of road reconstruction conducted in Zhangjiagang, Suzhou, China. Steel slag was used in the structural layers to enhance the water permeability of the pavement and the removal of runoff pollutants. The combined effects of this system and of the ribbon biological retention zone, allowed achieving an average removal rate of suspended solids, a chemical oxygen demand, a removal of total nitrogen and total phosphorus of 71.60%, 78.35%, 63.93%, and 49.47%; in contrast, a traditional road could not perform as well. Furthermore, the volume control rate of the annual runoff met the construction requirements (70%). The results of the present study indicate that, combining the traditional basic functions of roads with those of landscape and drainage might be a promising strategy for sponge city construction of urban road.

**Keywords:** urban water management; drainage function; permeable pavement; biological retention

## 1. Introduction

With the rapid process of urbanization and the increase of impervious surfaces in urban areas, great changes have taken place in the hydrological environment [1]. In recent years, urban point source pollution has been relatively controlled through continuous treatments and restoration. Urban non-point source pollution has gradually become a major problem for the improvement of the water environment, as it can be transported into flood flows from drainage systems [2,3]. Urban non-point source pollution derives from a complex dynamic process, which mainly refers to the scouring and carrying of surface pollutants by rainfall. Thus, the control of urban non-point source pollution is challenging. In order to control water pollution and improve the quality of the water environment and the utilization of rainwater, many countries have developed different concepts and technical measures according to their own conditions, such as measures based on low-impact development, best management practices, sustainable urban drainage systems, water-sensitive urban designs [4–6]. Considering this background, the sponge city (SC), a concept involving a series of innovative ideas, parameters, and methods, was proposed [7]. The SC emphasizes the usage of engineering and non-engineering measures to realize the accumulation, penetration, and purification of rainwater in urban areas.

Since the proposal of the SC concept, construction methods and models have been researched and actively explored in many cities [8,9]. This has allowed the integration of the concept of SC into many engineering construction fields such as architecture, park and green spaces, water systems, and urban roads [10]. As a special kind of land-use and an important part of the city, urban roads are the main transportation space and the embodiment of the urban landscape. The SC construction of urban roads (SCCUR) can not only achieve the goal of rainwater control, but also build a good platform for technology application and concept publicity [7,8]. However, since priority is given to urban roads' traffic function without considering the relationships between road, landscape, drainage, and other related environmental aspects, the SCCUR concept is rarely taken into account in practical engineering application. Thus, new models and technical strategies are needed for SCCUR.

In the traditional drainage model of urban roads, road runoff rainwater rapidly flows to the gutter inlet and then is discharged by a neighboring rainwater pipe. During this process, the road takes less time to discharge the runoff rainwater. As a result, substances such as suspended solids (SS), total nitrogen (TN), and total phosphorus (TP) may be transferred into the natural water body by runoff, which may cause serious pollution of natural water body and increase the chemical oxygen demand (COD). Thus, the traditional drainage model of urban roads had some disadvantages as reported below.

(1) Serious runoff pollution. As a special area carrying traffic, the runoff pollution of urban roads is serious [11]. With the scouring effect of rainfall, surface pollutants are carried into the water body. Using SC facilities to collect the runoff rainwater and remove the pollutants in rainwater is a critical approach for regional water environment improvement. (2) Large drainage pressure. The density of urban road networks increases rapidly with the development of urbanization, resulting in high watermark and poor soil permeability. After raining, a surface runoff will quickly form and enter rainwater pipes. Areas with an old rainwater system will face a large drainage pressure, which may lead to poor drainage, road ponding, and waterlogging. Using SC facilities is possible to collect the source runoff rainwater and extend its discharge time. This strategy can not only alleviate the drainage pressure on a rainwater drainage system, but also facilitate the natural water replenishment of plants [10]. (3) Poor traffic experience. Urban roads include motor lanes, non-motorized lanes, and sidewalks, which are usually impermeable [8]. In the traditional drainage system, rainwater inlets are used to collect road rainwater, and the interval between the rainwater inlets is generally 30–50 m. After long usage, the pavement surface of the sidewalks becomes loosen, and slight deformations appear in non-motor vehicle lanes, which causes ponding on roads. Travel on sidewalks and non-motorized lanes becomes difficult in rainy days.

With these disadvantages, many difficulties appear for the implementation of SCCUR. (1) Limitation of green space. According to design standards [12] and statistical analyses of the greening rate of multiple existing roads, the current greening rate of urban roads is generally less than 30%. The green space of urban roads is composed of a middle zone, a side zone, and a certain space on its two sides. Due to their particular function, belt-like urban roads are greatly limited in green space. Thus, the effective use of limited green space is a difficulty for SCCUR. (2) Establishing a drainage system. In the process of implementing SCCUR, drainage facilities need to be included in multiple SC facilities to ensure drainage security. For facilities with regulation and storage functions, overflow systems are especially needed. The overflow system generally includes rainwater collection pipes in the drainage layer of the facilities, overflow inlets, and rainwater pipes connected with the conventional drainage system. In actual engineering projects, the drainage safety should be ensured, and two sets of drainage systems should be avoided, which requires the establishment of a complete system including an anterior collection system and a back-end discharge system. (3) Professional coordination. SCCUR mainly uses various technical SC facilities to collect the runoffs and reduce pollutants. These SC facilities are mainly constructed in roads and green space, but roads and greening are not the main scope of

drainage. Thus, these three features of road, landscape, and drainage need to be highly integrated during the actual process of SC construction.

Based on the above disadvantages, a new strategy for SCCUR with a drainage system as the core is proposed in the present study. Suitable SC facilities were constructed on a road, which was reconstructed using the proposed strategy. By analyzing the construction process and related achievements, the feasibility and key points of this strategy are discussed.

## 2. Materials and Methods

### 2.1. Project Overview

The present work was conducted on Liangfeng Road in Zhangjiagang, Suzhou, as a pilot project (Figure 1). The road was being remodeled for the improvement of streets and lanes in the old district within the construction of SC with the specific aim to improve road, landscape, and drainage. The length and width of the reconstructed site were about 400 m and 27–32 m, respectively, and a typical section is shown in Figure 1. The whole road space includes two-way motor lanes, non-maneuverable lanes, and sidewalks. The green space was scarce and concentrated at the end of the road. The main problems of this road were damage of the pavement, low green coverage, disorder of the drainage system.

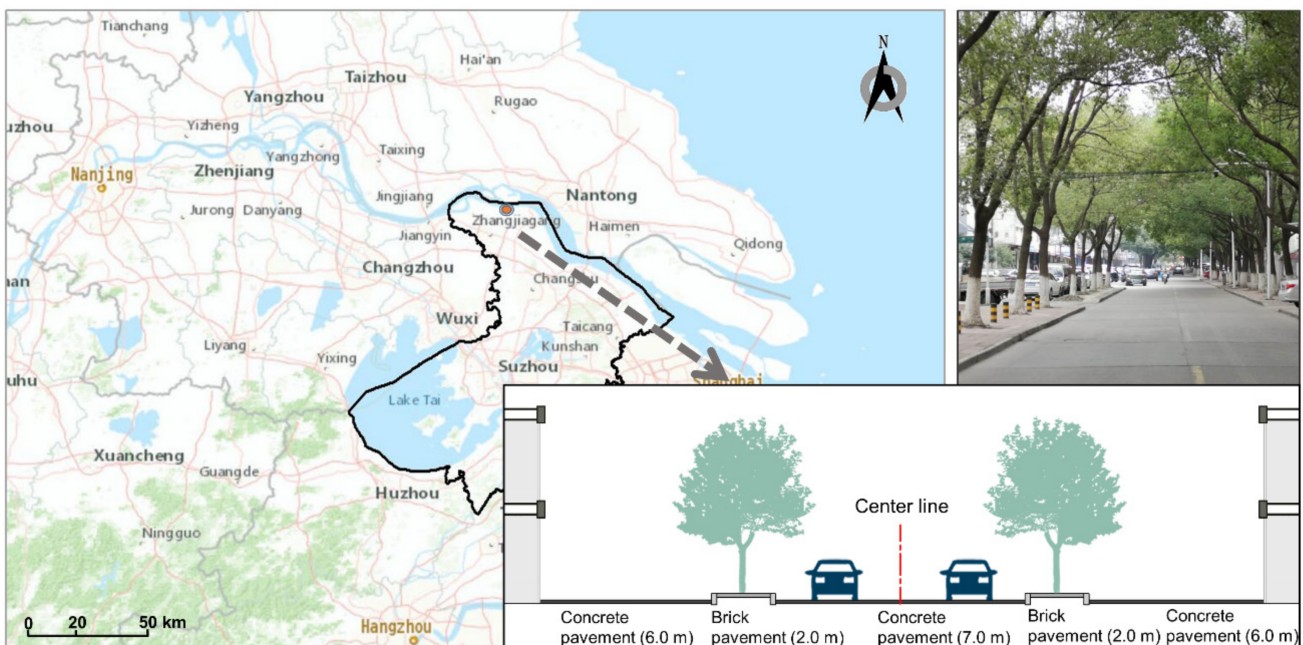

**Figure 1.** Location and typical section of the pilot road.

Facilities have become diversified with the development of SC. The main functions of SC facilities are infiltration (permeable pavement, sunken green space, biological retention, infiltration well), storage (wet pond, stormwater wetland), transfer (grass filter, infiltration pipe, infiltration channel), and purification (vegetation buffer zone, initial rainwater disposal facility). The selection of SC facilities should be based on the use of each road partition, and an optimization design should be realized according to the site conditions on the basis of combining the actual construction demands. The parameters should be defined pertinently to ensure the efficiency of the SCCUR.

### 2.2. Design Scheme

The process of SCCUR involves different features, such as road, landscape, and drainage. SCCUR needs to build a drainage system including the collection, treatment, and discharge of the runoff rainwater. Thus, a comprehensive consideration of the slope, elevation, and plant configuration of the urban road is necessary. After analyzing the construction conditions of the urban road and formulating a preliminary SC scheme, the

construction of different features should be coordinated, with the drainage system as the core. The main aspects of different features and the used integration methods are presented in Figure 2a.

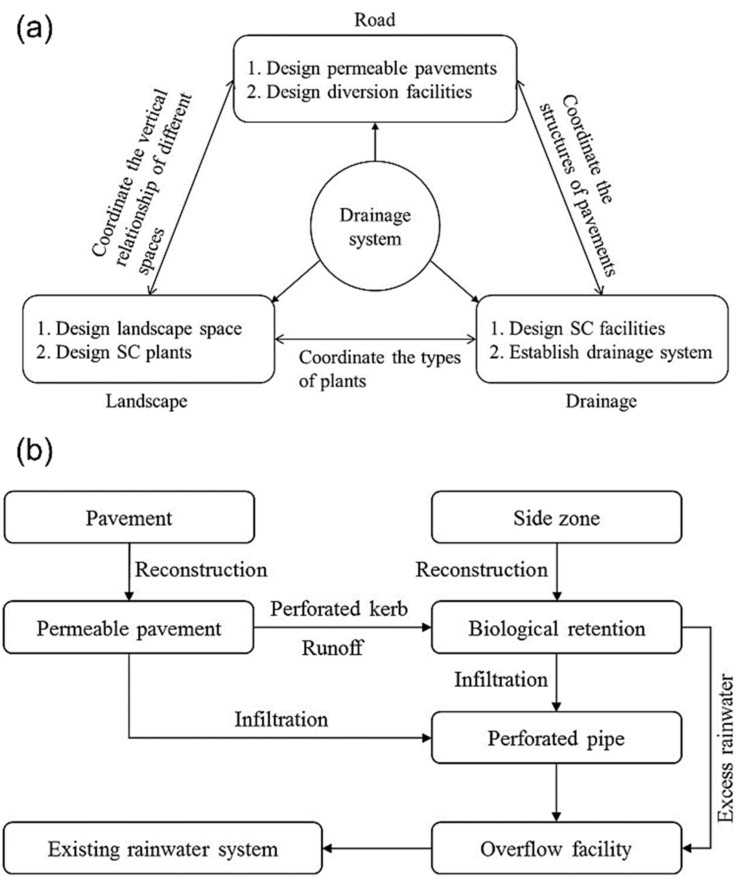

**Figure 2.** (**a**) Main aspects of different features in the design of SCCUR; (**b**) transformation strategies for different surfaces.

The vertical and transverse of the pavement need to be optimized while reconstructing the pavement, which plays a critical role in the collection of runoff rainwater. Due to the construction conditions (schedule, investment, etc.), the renewal of the motor lane involved paving asphalt concrete on the existing concrete pavement. To achieve the goals of this reconstruction, the pavement of the non-motor lane and sidewalk were planned to be demolished and rebuilt from foundation to surface. The street trees of this road were located on the sidewalk, which had a serious impact on the traffic function of the sidewalk. In order to improve the green space and optimize the traffic function of this road, we decided to transform the sidewalk into green space and divide the sidewalk in the outer space of the non-motor lanes. The SC scheme of this road and the transformation strategies of different surfaces are shown in Figure 2b.

### 2.3. Sampling and Analytical Methods

The reduction of runoff pollution is mainly realized through the comprehensive action of various SC facilities in the construction of SC. Research shows that SC facilities (e.g., permeable pavement, biological retention, etc.) studied in the laboratory can effectively reduce a variety of pollutants. In order to further analyze the effect of SCCUR, the reduction capacity of natural runoff pollution of SC facilities constructed in this case was studied.

According to the actual construction situation under exam, we studied the pollutants removal performance of steel slag–permeable asphalt pavement (fully pervious) and ribbon biological retention zone (RBRZ) by analyzing the pollutant indexes of water samples.

The sampling locations were selected based on the guidelines of different sampling techniques [13]. Generally, the locations were the typical cross sections of the road, and the sampling points were water inlet, overflow outlet, and outlet of porous drainpipes [14]. Multiple samples were collected with the random points method on the typical cross sections of the road. These water samples were collected by self-made devices, and the collection points of effluent water were at the end of the pipes in the drainage layer of steel slag–permeable asphalt pavement and RBRZ. The water samples of the control group were collected at the rainwater inlet on the section of the pilot road not undergoing reconstruction. The pollutants in urban runoff rainwater are generally subjected to the first flush phenomenon, and the collection of surface runoff rainwater in the initial stage of rainfall is particularly important [15]. The sampling interval time increased gradually with the progress of rainfall, and the sampling times were 5, 10, 15, 20, 25, 30, 45, 60, 90, 120 min after the formation of runoff. The water samples were collected in 500 mL polyethylene bottles and labeled. All collected water samples were immediately transported to the laboratory and subjected to analysis within 48 h.

Influent and effluent water samples were tested for water quality parameters (SS, COD, TN, TP). Water quality was monitored by measuring SS, COD, TN, and TP with standard procedures [16]. Generally, SS was measured with the gravimetric method, COD was measured with potassium dichromate titration, TN was measured with potassium persulphate digestion–UV spectrophotometry, and TP was measured with ammonium molybdate spectrophotometry.

## 3. Results

### 3.1. Establishment of a New Model of SCCUR

Figure 3a shows the typical runoff organization and drainage model of urban roads. Although this model can rapidly collect and discharge the runoff, it also carries a large amount of pollutants to the water body. The pollutants brought by runoff are important contributors to water pollution. The key of our new strategy is a change in the mode of controlling runoff rainwater. The control of the runoff and the removal of the pollutants can be realized by optimizing urban roads and green spaces and by building suitable SC facilities.

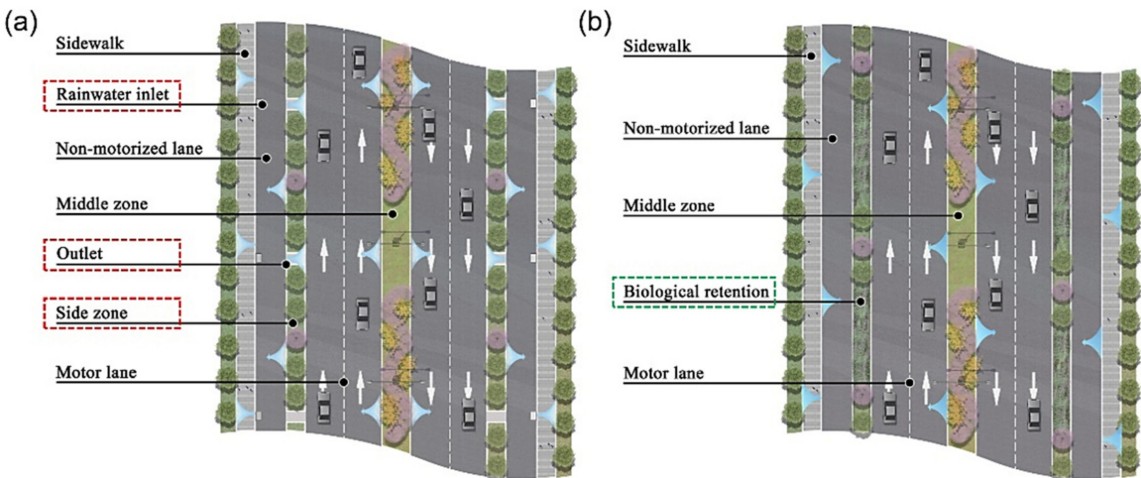

**Figure 3.** (**a**) Typical runoff organization and drainage model of urban roads; (**b**) optimization of runoff rainwater.

The conditions of SC construction in areas of urban roads should be analyzed based on the actual construction requirements. For reconstructed roads, functional requirements (pavement repair, etc.) and non-functional requirements (activity space optimization, etc.) should also be analyzed. A reasonable scheme of the SCCUR should be formulated on the basis of thorough investigations.

In order to realize the runoff control of urban roads, the conventional mode of runoff should be changed (the red dash boxes marked in Figure 3a). A new strategy should not only avoid runoff entering the rainwater pipes rapidly, but also avoid road ponding. This goal can be achieved through two measures: by improving the water permeability of the pavement and delaying the formation of runoff, and by setting up SC facilities with rainwater regulation, storage, and purification capacity to collect and purify rainwater. In the new strategy of SCCUR, the vertical and transverse design of roads and green spaces should be considered comprehensively. The optimized strategy is shown in Figure 3b.

*3.2. Pilot Study*

3.2.1. Background of the Site

The site was selected at Suzhou (Figure 1). In combination with the construction objectives and engineering documents, the site was surveyed, and a preliminary scheme of the SCCUR was formulated. The construction objectives of the SCCUR generally include several aspects, and the specific objectives need to be comprehensively analyzed according to the requirements and demands of the project location. According to these, the main problems of this project were analyzed, and solutions were proposed. Regarding the road, the concrete pavement was cracked, and the sidewalk could not properly allow walking. Regarding the landscape, the green space was limited, and the green layer was unvarying. Finally, the drainage system of this road had been built for a long time without meeting the standard requirements. Ponding also occurred in this road. With these problems, the following solutions were proposed: realization of a permeable pavement, optimization of the spatial layout to improve the green coverage rate and enrich the green layer, and use of several SC facilities to optimize the organization of the runoff and relieve the pressure on the rainwater pipes. A comprehensive design with the drainage as the core was formulated to eliminate road waterlogging and reduce the pollution of the runoff.

Based on the analysis of the requirements, objectives, and conditions of this road, the major features to integrate in the reconstruction were determined to be the road, the landscape, and the drainage. An SC scheme of this road with the drainage system as the core was formulated to renew the pavement, improve the landscape, and optimize the drainage system.

3.2.2. Design of the SC Facilities

In order to further study the application of steel slag and the performance of the steel slag–permeable asphalt mixture in the construction of SCCUR, two structures of steel slag–permeable asphalt pavement were designed for the renewal of the motor lane and non-motor lane. The structures of the steel slag–permeable asphalt pavement are shown in Figure 4a,b. The structural layers were different between the semi-pervious and the fully pervious structures. Steel slag and more layers were designed in the fully pervious structure for the water permeability of the pavement. For the steel slag–permeable asphalt mixture, the production mixture proportion and the production gradation of steel slag are shown in Tables 1 and 2, respectively. It should be noted that the whetstone ratio (weight ratio between oil and stone) was 4.5%, as shown in the Table 1. The main steel used in the present study was from 3# and 4# hot bin, with steel slag size of 6–15 mm. A larger size (greater than 9.5 mm, Table 2) of the steel slag was favorable for the drainage.

The structural layers of the steel slag–permeable asphalt pavement consisted of steel slag–permeable brick, screed coat, steel slag-pervious concrete, and a gravel drainage layer (Figure 4c). With these layers, the steel slag–permeable asphalt pavement acquired a perfect water permeability, allowing rainwater to infiltrate through the structural layers and finally enter the rainwater system through a pipe in the gravel drainage layer, which had 1% openings for water collection.

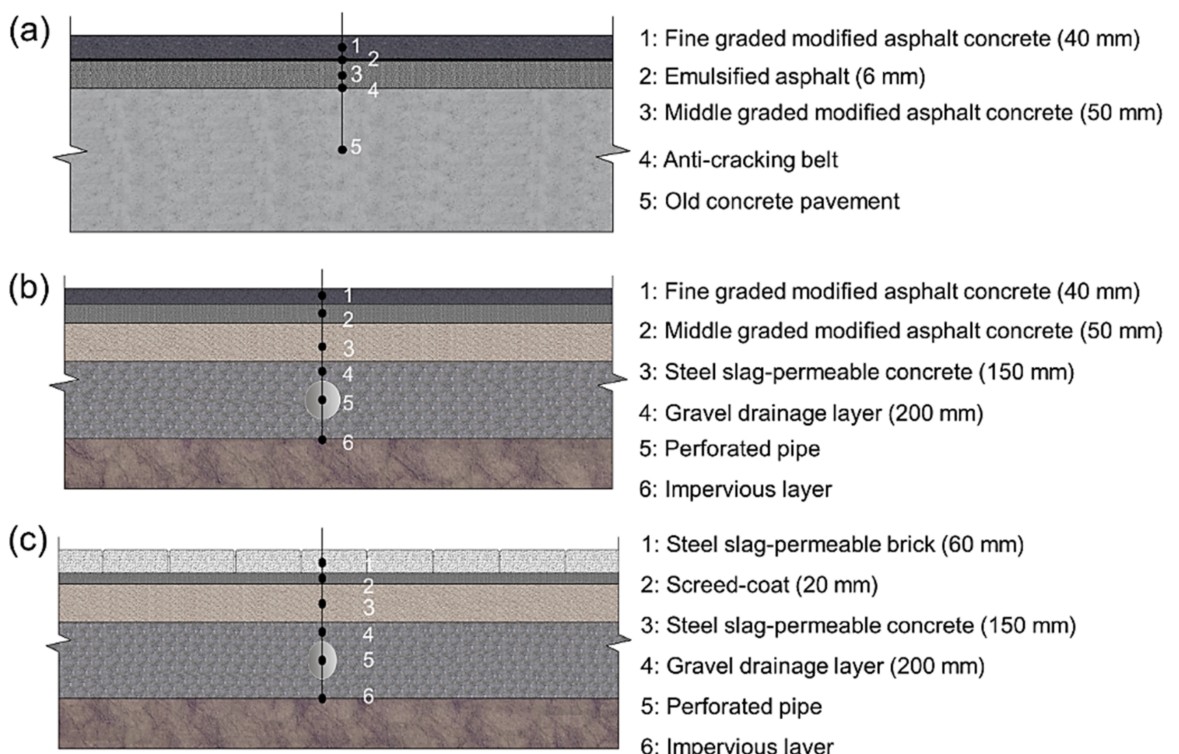

**Figure 4.** Structures of the steel slag–permeable asphalt pavement: (**a**) semi-pervious; (**b**) fully pervious; (**c**) structures of the steel slag–permeable concrete pavement (fully pervious).

**Table 1.** Proportion of steel slag in the mixture.

| Hot Material Warehouse (Size, mm). | Proportion (%) |
|---|---|
| 1# (0–3) | 11.5 |
| 2# (3–6) | 0 |
| 3# (6–11) | 49.0 |
| 4# (11–15) | 34.5 |
| Mineral powder | 5.0 |

**Table 2.** Production gradation of the steel slag.

| Diameter of Square-Opening Sieve (mm) | Mass Proportion of Sifting (%) | |
|---|---|---|
| | Design Gradation | Design Standard |
| 16 | 100 | 100 |
| 13.2 | 98.1 | 90–100 |
| 9.5 | 60.8 | 50–80 |
| 4.75 | 16.8 | 12–30 |
| 2.36 | 13.7 | 10–22 |
| 1.18 | 11.3 | 6–18 |
| 0.6 | 9.2 | 4–15 |
| 0.3 | 8.0 | 3–12 |
| 0.15 | 7.1 | 3–8 |
| 0.075 | 5.1 | 2–6 |

Based on the above design, the steel slag–permeable asphalt mixture was prepared, and performance tests were conducted. Leakage loss, dispersion loss, and Marshall residual stability were analyzed, and the results are reported in Table 3. On the basis of the standard technical requirements e, the steel slag–permeable asphalt mixture had a good performance, especially as concerns its stability and permeability (Table 3).

**Table 3.** Performance test results of the steel slag–permeable asphalt mixture.

| Test Items | Detection Value | Technical Requirements |
|---|---|---|
| The number of actual hits of the Marshall specimen | 50 | - |
| Whetstone ratio (%) | 4.5 | - |
| Relative density of gross volume | 2.421 | Measured |
| Theoretical relative density | 3.036 | Calculation |
| Porosity (%) | 20.3 | 20–22 |
| Stability (KN) | 9.86 | $\geq$5.0 |
| Binder loss of Schellenberg asphalt leakage test (%) | 0.22 | $\leq$0.8 |
| Mixture loss of the Fort Kentucky flying test (%) | 7.0 | $\leq$15 |
| Mixture loss of the water immersion Kentburgh scattering test (%) | 14.6 | 20 |
| Residual stability in immersion Marshall test (%) | 92.8 | $\geq$85 |
| Permeability coefficient (mL/min) | 5433 | $\geq$5000 |

### 3.2.3. Ribbon Biological Retention Zone

Considering the actual conditions of the site under exam, the structural layers of the RBRZ included an aquifer (150 mm), a planting soil layer (400 mm), a filter layer (300 mm), and a gravel drainage layer with perforated pipes (250 mm). A permeable geotextile was laid between the different structural layers to ensure pollutant removal and stability of RBRZ. The drainage safety was ensured by the perforated drainage pipes in the gravel layer, and a fabric protecting against osmosis was laid under the gravel layer to ensure the safety of the subgrade. The values of the pivotal design parameters of the RBRZ are shown in Table 4. Among these parameters, planting soil plays a key role in pollutant removal, while filter layer and gravel drainage layer favor permeability.

**Table 4.** Design parameters of the RBRZ.

| Structural Layer | Minimum Permeability Coefficient K (m/s) | Minimum Void Ratio (%) | Material Specifications |
|---|---|---|---|
| Planting soil | $1.5 \times 10^{-5}$ | 5 | Evenly mix 45% medium sand, 10% pine bark, 5% nutrient soil |
| Filter layer | $10 \times 10^{-5}$ | 10 | Ceramsite matrix of $\varphi$ 7–10 mm |
| Drainage layer | $100 \times 10^{-5}$ | 15 | $\varphi$ 20–30 mm gravel with perforated drainage pipe |

### 3.2.4. Diversion Facilities

The key of the SCCUR is to ensure the runoff rainwater can effectively enter the SC facilities with water storage capability, such as the RBRZ. For the purpose of traffic safety, kerbs are generally set between the pavement and the green belts of urban roads (Figure 5a,b). Several kerbs were perforated to ensure that the runoff rainwater formed on the pavements could enter the RBRZ. The kerbs were made of granite, with size of 99 cm $\times$ 30 cm $\times$ 15 cm.

Based on practical considerations and requirements, two holes with size of 30 cm $\times$ 7.5 cm were made in each kerb. The distance between the perforated kerbs was 15 m, and the kerbs were fixed with concrete. The vertical relationship between pavement, perforated kerb, and RBRZ is shown in Figure 5c.

Overflow inlets were set in the RBRZ for drainage safety, so to allow the excess rainwater in the RBRZ to enter the drainage system through the overflow inlets, when rainfall exceeds the water storage capacity of the RBRZ. As shown in Figure 5d, the used overflow inlets in the shape of a platform have five inlet surfaces, which significantly reduces the risk of blockage. The maximum overflow capacity of the overflow inlets with a size of 75 $\times$ 45 cm was 30 L/s, and the interval between them was 25–30 m.

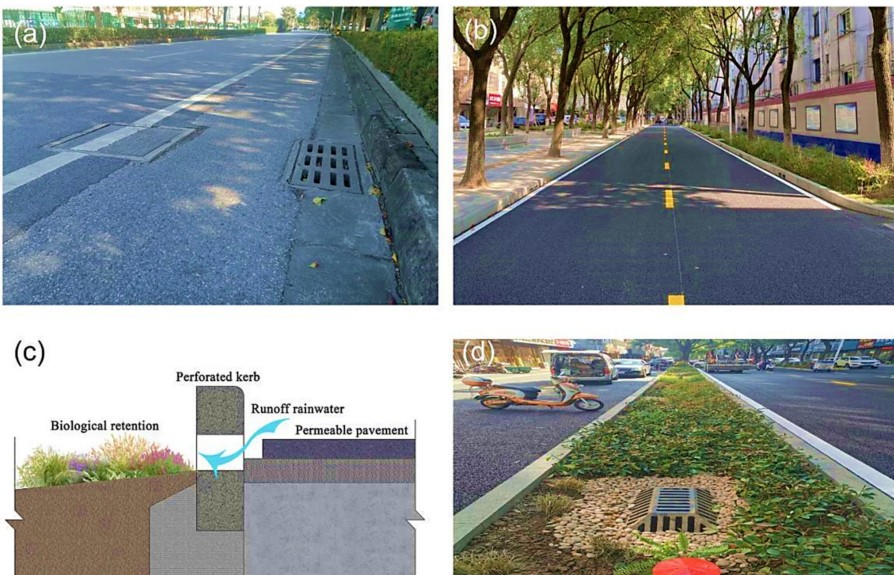

**Figure 5.** The road before (**a**) and after (**b**) reconstruction; (**c**) vertical relationship between different parts; (**d**) schematic diagram of the overflow inlet.

### 3.2.5. Runoff and Pollution Control Capacity

The runoff volume control capacity of this road was checked based on the reconstruction scheme, using Equations (1)–(4) [17,18].

$$V_w = A_f h_m (1 - f_v) \tag{1}$$

where $V_w$ (m$^3$) is the water storage volume, $A_f$ (m$^2$) is the area of the sampling location, $h_m$ (m) is the maximum storage height (<0.3 m), $f_v$ is the proportion of the plant cross-sectional area to the water storage area (0.15–0.3);

$$G = A_G \left( n_1 d_{f1} + n_2 d_{f2} \right) \tag{2}$$

where $G$ (m$^3$) is the water storage of the internal structure, $A_G$ (m$^2$) is the area of the facility area, $n_1$ is the average porosity of the planting soil, $d_{f1}$ (m) is the depth of the planting soil, $n_2$ is the porosity of packing layer, $d_{f2}$ (m) is the depth of the filler layer;

$$V = 10 H \Psi F \tag{3}$$

where $V$ (m$^3$) is the water storage, $H$ (m) is the depth of the design rainfall, $\Psi$ is the rainfall runoff coefficient, $F$ (m$^2$) is the service area;

$$R_w = \frac{V_w + G}{V} \tag{4}$$

where $R_w$ is the water volume control rate.

For the examined location, the values of the $A_f$, $h_m$, and $f_v$ were 4, 0.15, and 0.2; thus, $V_w$ resulted to be 0.48. The values of the $A_G$, $n_1$, $d_{f1}$, $n_2$, and $d_{f2}$ were 4, 0.05, 0.4, 0.12, and 0.55; thus, $G$ was calculated as 0.35. The values of the $H$, $\Psi$, and $F$ were $19.4 \times 10^{-3}$, 0.65, and 23; thus, $V$ was calculated as 0.29. Based on the above results, $R_w$ resulted to be 2.86. This value was greater than that in a previous report (1.32) [19], which indicated that we obtained a greater runoff volume control capacity.

Event mean concentration analysis is used as the standard to evaluate the runoff quality; it has a high confidence level in the comparison between different catchment areas. The average event means the concentration removal rate of SS, the COD, and the removal rates of TN and TP by the steel slag–permeable asphalt pavement (fully pervious) were

61.8%, 66.84%, 56.33%, and 47.36%; the results for RBRZ were 71.60%, 78.35%, 63.93%, and 49.47% (Figure 6). The removal rate was comparable to that in previous reports [20,21]. For example, for porous asphalt–bio-retention combined roads, the removal rates of SS, TN, and TP were 70. 26%, 46. 29%, and 19. 27% [20].

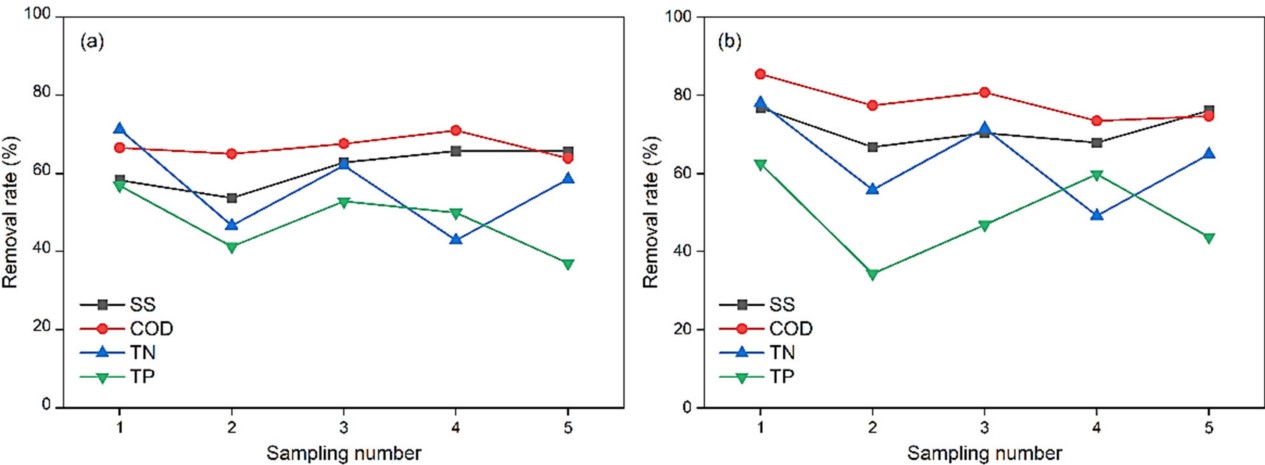

**Figure 6.** Removal rate of SS, COD, removal rates of TN and TP of the steel slag–permeable asphalt pavement (**a**) and the ribbon biological retention zone (**b**) in different samples.

## 4. Discussion

As a strategy of stormwater management, SC has many advantages. In terms of urban roads, the construction goals mainly include the following aspects: 1. The volume control rate of annual runoff. In order to meet the requirement of the volume control rate of annual runoff, various suitable facilities can be used to store the runoff rainwater. These SC facilities can meet the requirement that a certain amount of runoff will not be discharged. 2. Removal of runoff pollutants. The interception and filtration functions of these facilities can be used to reduce the pollutants in the runoff rainwater, so as to reduce the total discharge of rainwater pollutants of the site. 3. Integration of different functions. With the development of cities, the functional requirements of urban roads have gradually changed from simplification to diversification. The SCCUR needs not only to control the runoff and remove pollutants, but also to integrate the requirements of road, landscape, and drainage, thus creating a multi-functional road space while realizing road renewal.

In the present SCCUR, several SC facilities were constructed, and the runoff organization was optimized to reduce the total amount of discharged rainwater. In the examined case, the volume control rate of annual runoff needed to approach 70%, and the dependable rainfall was 19.4 mm. The removal of pollutants (SS, TN, TP, etc.) and a proper COD in runoff rainwater is another objective in road reconstruction, and the degree of SS removal is generally used as a typical pollutant index in SCCUR due to its certain correlation with other pollutant indexes [22]. In this case, the comprehensive removal rate of SS in the runoff needed to approach 50% after the reconstruction. Furthermore, with the development of the city, urban roads are endowed with more functions. Therefore, a road needs to be built into an ecological block integrating traffic, recreation, and walking on the basis of runoff control and road renewal.

In particular, a permeable pavement, a typical type of SC facilities, can greatly improve hydrology and water quality. With the development of pavement engineering, the types of permeable pavements suitable for various underlying surfaces are gradually increasing in number. In many occasions, permeable pavements can not only meet their functional requirements, but also improve runoff control and runoff pollution reduction compared to impervious pavement [23]. As an alternative to traditional asphalt pavements, permeable asphalt pavements are generally used in traffic roads, parking areas, etc. Permeable asphalt pavements have excellent performance in drainage, antiskid function, and noise reduction.

When producing permeable asphalt pavements, structural competition, connected air voids, and aggregates should be selected and designed systematically according to the rainfall characteristics, as they have a critical impact on permeability and bearing capacity [24].

However, natural aggregate resources are facing the problem of overexploitation; the incorporation of recycled aggregate in asphalt mixtures is an efficient method to preserve resources [25]. As an industrial solid waste with the largest output, steel slag has the characteristics of large porosity, high hardness, and good particle shape. Compared with ordinary asphalt mixtures, the steel slag–permeable asphalt mixture has better performance as regards repeated fluctuation under low temperatures, the snow melting process on an electrical-thermal pavement system, permeability, and water stability [26,27]. In the present study, steel slag was applied. The results of the runoff volume control capacity showed that the above parameters met the requirements of relevant standards. The appearance of the paved steel slag–permeable asphalt pavement was uniform and flat, without segregation and oil spots, and the performance met the design requirements. These results suggest a new way of steel slag utilization. More in-depth research can be carried out in future practical projects.

After optimizing the spatial layout, the sidewalk was moved to the outside of the non-motor lane, located on the outermost side of this road. In order to improve the travel experience in rainy days and meet the requirements of the SCCUR, the steel slag–permeable concrete pavement was placed on the sidewalk. Although permeable pavements have a positive effect on runoff coefficient reduction and runoff pollutants interception, their storage capacity of rainwater is limited. The slope of the permeable pavements should be designed considering other facilities, so that rainwater can enter those facilities with water storage capacity when it exceeds the infiltration capacity of the permeable pavements.

Biological retention (bioretention) is a kind of SC facility that can be subdivided into many types according to its functions and use. Bioretention has a good removal effect on pollutants in runoff, and the design of parameters (such as the type of filter material, the thickness of the aquifer, etc.) has a certain impact on its removal capacity [28,29]. In the present study, after the demolition of the brick sidewalk (2 m wide), the space was restored to green belt. According to the demands of rainwater storage and purification, these green belts needed to be sunk.

Furthermore, in order to control rainwater of this road, the runoff formed on the pavements was diverted to the RBRZ, and the number of rainwater inlets on this road was reduced markedly for better implementation. For analyzing the SCURR achievements, the volume control rate of annual runoff (not less than 70%) was verified. The SC facility can fully control the discharge of runoff rainfall within the design range when its water volume control rate is $\geq 1$. The results of runoff control capacity show that the rainwater in the service of RBRZ could not be discharged directly when the rainfall was 19.4 mm, which met the construction requirements of this road (for which the volume control rate of annual runoff was 70%). Due to the fact that the plants in the RBRZ occupied a certain storage space, a reduction factor of water storage capacity had to be considered during the of verification. The effective space of the aquifer in this case was calculated as 80% based on the different plants.

As runoff is a complex hydrological process, pollutant concentration is easily affected by rainfall, runoff, and other factors. Thus, event mean concentration was used to benchmark the performance of pollutants removal in this study [30]. As shown in Figure 6, both steel slag–permeable asphalt pavement (fully pervious) and RBRZ constructed in this pilot work had a significant removal capacity of conventional pollutants in runoff rainwater. The RBRZ showed a higher removal capacity than the steel slag–permeable asphalt pavement (fully pervious), which was likely due to the specific removal mechanism of the RBRZ.

## 5. Conclusions

Owing to the limited service area of point facilities, it is difficult that the traditional drainage mode of urban roads can meet the requirements of SC; therefore, in this case, the effect of SCCUR is insignificant. A new construction strategy with the drainage system as the core was proposed in this paper. The key findings are as follows:

1.  New design strategy. The construction strategy focused on the organization of runoff rainwater in the road space, which could be achieved through vertical optimization of the structure. Thus, the vertical relationship of different facilities needs to be designed in an integrated way, and the basic functions of the facilities need to be met. While ensuring the drainage safety, the runoff rainwater shall be diverted to the nearest SC facilities with capacity of retention and pollutant removal.

2.  Additional advantages of the new strategy. Generally, the road and the green space were used to create a sponge road combining basic road functions with landscape effect and drainage capacity. The reconstruction of a road in Zhangjiagang with this new strategy was successful. Through the construction of permeable pavements, biological retention facilities, and other facilities, optimization of runoff organization, volume control of the runoff, removal of pollutants, and renewal of the road were realized. The construction of SC not only can achieve the goal of stormwater management, but also is an important way to realize urban renewal.

Therefore, the combined usage of SC facilities has a significant effect on the control of runoff rainwater pollution according to the effective pollutant's removal performance of the reconstructed road. The selection and design of facilities shall be based on the characteristics of runoff pollutants, and the construction quality should be strictly controlled. These results support the design and construction of SC urban roads.

However, there are some limitations in the present study. For example, the composition of steel slag was not analyzed, the design of the steel slag–permeable asphalt pavement was not optimized, and the runoff pollution process was not investigated. In a further study, the optimization, decontamination mechanism, effect of climate change, and cost evaluation should be considered. With the above information, the methods in the present study could be largely adopted.

**Author Contributions:** Conceptualization, C.W. and T.H.; methodology, C.W., J.W. and P.L.; validation, H.L. and T.H.; investigation, C.W., H.L. and Y.J.; data curation, C.W. and B.W.; writing—original draft preparation, C.W.; writing—review and editing, T.H.; visualization, C.W. and B.W.; supervision, T.H.; project administration, T.H.; funding acquisition, T.H. All authors have read and agreed to the published version of the manuscript.

**Funding:** This research was funded by Key Laboratory of Suzhou Sponge City Technology (Grant No. SZS2021265), the Suzhou Science and Technology Plan Project—Minsheng Science and Technology (Suzhou science and Technology Bureau, Grant No. SS202002), the Water Pollution Control and Treatment, National Science and Technology Major Project (Grant No. 2017ZX07205002).

**Acknowledgments:** We are grateful to Xiaoyi Xu and Wei Wu of Suzhou University of Science and Technology for insightful discussions.

**Conflicts of Interest:** The authors declare no conflict of interest.

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
