# Peer review of "A New Strategy for Sponge City Construction of Urban Roads: Combining the Traditional Functions with Landscape and Drainage"

_water, doi:10.3390/w13233469_

Round 1

Reviewer 1 Report

ABSTRACT:

English is extremely poor and needs to be revised. The abstract contains very long sentences that need to be shortened. Furthermore, it needs a bit more clarification about the need of the work, the purpose, the methods. Authors have added a hint of the results which is great and can attract the interest of the reader.

INTRODUCTION:

Considering the topic of the paper, the following manuscript should be definitely be included within the literature review because it is extremely relevant for Sponge Cities (should go with references in line 38):

Matteo Rubinato, Andrew Nichols, Yong Peng, Jian-min Zhang, Craig Lashford, Yanpeng Cai, Pengzhi Lin, Simon Tait, Urban and river flooding: Comparison of flood risk management approaches in the UK and China and an assessment of future knowledge needs, Water Science and Engineering,Volume 12, Issue 4, 2019, Pages 274-283, ISSN 1674-2370, https://doi.org/10.1016/j.wse.2019.12.004.

Furthermore, considering the main topic is the increase of pollution in urban areas, authors should also include the following manuscript because it deals with the problem of pollutant transport during flooding events, therefore it needs to be incorporated (should be as a reference in line 30):

Md.N.A. Beg, M. Rubinato, R.F. Carvalho, J. Shucksmith. CFD modelling of the transport of soluble pollutants from sewer networks to surface flows during urban flood events. Water, 12 (9), 2514, https://doi.org/10.3390/w12092514 

Which are the design standards you are referring to in lines 81-82? There should be a reference. There should be also some comparisons with standards across other countries, to enhance the critical aspect for the design which is missing here at the moment.

MATERIALS AND METHODS

In figure 1 it is impossible to read the text in the scheme on the bottom left

Table 1 is extremely poor and could be simply be presented in text or bullet points.

Figure 2 should be bigger to make it easier to be read.

Where multiple samples collected in similar areas? Is there a grid for the measurements or where they random points?

How where the sampling locations selected?

More details about the standard procedures for the measurements should definitely be listed.

Again Table 2 has a very poor format. Please revise.

What was used to calculate the runoff? Authors should also test increments expected due to climate change (higher intensities and higher numbers of impermeable surfaces).

Results should be compared with more existing studies available in literature.

CONCLUSIONS

Outcomes are listed properly however you should expand the statements about applications of these results and the implications to adopt these methods for large scales.

Reviewer 2 Report

The topic of the paper under the title “A New Strategy for the Sponge City Construction of Urban Roads: Combines the Traditional Ability with Landscape and Drainage” is very interesting, current and within the scope of the Water journal. The manuscript is quite well written. However, some issues require clarification and improvement. Therefore, I recommend major revisions before the publication of the manuscript. Please find some details below.

  1. My main concern is using lumped references. In the opinion of the reviewer, the authors abuse this form of citation. This is true for both the introduction (lines 38, 44, 62) and the discussion (lines 309, 342). The lumped references also appear in the Materials and Methods section (lines 152, 162). For example, in line 38, the authors cite as many as seven items in one place, without giving any details about the referenced articles. It looks as if the authors want to cite as many publications as possible without taking the time to analyze them in depth. Please avoid using this form of citation. No more than three items should be cited in one place. The indicated fragments of the text should be expanded to emphasize what exactly was contained in the referenced articles.
  2. I also have reservations about dividing the manuscript into subsections. For example, subsections 3.2.2.1 and 3.2.2.2 are very short. Is this division necessary? The other subsections also contain very little text. I understand that the discussion has been included in the next section, but tables and figures require more comment in section 3.
  3. The last section should describe the limitations of the research and the directions of further research.

Please also consider the following issues:

  • The keywords should be different from the words in the title. Please remove or change the following keywords: “sponge city”, “urban road”.
  • Please present Table 1 in a different form. It is currently unreadable.
  • Lines 170-171 – “Analytical Methods for Water and Wastewater Monitoring” – Please add this item to the References.
  • The title of Table 3 should be on the same page as Table 3. It is now on page 6, while the table is on 7.
  • The reference to Table 5 should be placed in the text before Table 5. It is currently only on page 10.
  • Line 270 – There are no spaces before the table number.
  • Line 350 – No dot before "7" is needed.
  • The 11th item of the References has a different formatting.
  • References 17, 18 and 19 do not have numbers on the References list.

Best regards

Reviewer 3 Report

See attached.

Round 2

Reviewer 1 Report

Thanks for addressing the comments provided.

Reviewer 2 Report

Thank you for considering my comments. In my opinion, the paper can be published.